# ABC Transporters in Bacterial Nanomachineries

**DOI:** 10.3390/ijms24076227

**Published:** 2023-03-25

**Authors:** Florestan L. Bilsing, Manuel T. Anlauf, Eymen Hachani, Sakshi Khosa, Lutz Schmitt

**Affiliations:** Institute of Biochemistry, Heinrich Heine University, Universitätsstr. 1, 40225 Düsseldorf, Germany

**Keywords:** ABC transporter, nanomachineries, Mac system, Lpt system, Mla system, Lol system

## Abstract

Members of the superfamily of ABC transporters are found in all domains of life. Most of these primary active transporters act as isolated entities and export or import their substrates in an ATP-dependent manner across biological membranes. However, some ABC transporters are also part of larger protein complexes, so-called nanomachineries that catalyze the vectorial transport of their substrates. Here, we will focus on four bacterial examples of such nanomachineries: the Mac system providing drug resistance, the Lpt system catalyzing vectorial LPS transport, the Mla system responsible for phospholipid transport, and the Lol system, which is required for lipoprotein transport to the outer membrane of Gram-negative bacteria. For all four systems, we tried to summarize the existing data and provide a structure-function analysis highlighting the mechanistical aspect of the coupling of ATP hydrolysis to substrate translocation.

## 1. The Classical View of an ABC Exporter

ATP-Binding Cassette (ABC) transporters are present in all domains of life, from prokaryotes to eukaryotes [1]. ABC exporters consist of two nucleotide-binding domains (NDBs) and two transmembrane domains (TMDs) [2,3]. The NBDs are responsible for binding and hydrolyzing ATP, and the TMDs allow substrate translocation across the membrane [4]. The conformational changes of the NBDs, due to ATP binding and hydrolysis, are translated via coupling helices to the TMDs [5,6]. While the NBDs are highly conserved among all ABC transporters, the TMDs show a huge variety that goes in line with the huge variety of ABC transporter substrates [7]. Within this review, we will focus on prokaryotic ABC exporters functioning in nanomachineries. Some of the prokaryotic ABC exporters are “half-size” transporters with one NBD and one TMD fused together on a single polypeptide chain, while most harbor NBDs and TMDs as separate polypeptide chains allowing heterooligomeric assemblies (see Figure 1) [3].

Without any structural information, Jardetzky et al. proposed already in 1966 two configurations for a membrane transporter enabling a ‘two-sided access’ to a central cavity within the transporter [8]. Today we know from multiple protein structures that most of the ABC exporters change during substrate translocation from an inward-facing (IF) state to an outward-facing (OF) state [9,10,11,12,13]. This model of a translocation cycle (see Figure 2) was extended by the outward-occluded state observed for the peptide transporter McjD [14] and for other exporters as well [15]. New findings revealed even more transition states within one translocation cycle [16]. Besides this classical understanding of changing between IF and OF states, there are further models expanding the mechanisms for substrate translocation. On the one hand, there is the alternate access model, where a substrate is not entering from the cytoplasm but from a lateral opening in the TMDs (PCAT1 [17]). On the other hand, there is the outward-only model, where the substrate cavity is open and closing without an inward-facing state (PglK [18]). Similar to this outward-only model, simulations suggested a Constant Contact Model of the NBDs without a wide open IF state [19].

The power stroke for transmission from IF to OF state, and along with this, the substrate translocation, is supposed to be the binding of ATP. ATP hydrolysis, therefore, only resets the system back into the IF state [20,21]. Moreover, it is conceivable that there is a difference in the power stroke for heterodimeric ABC exporters that contain an intrinsically impaired nucleotide binding site, as it is proposed for BmrCD. Here, turnover from IF to OF state is exclusively attributed to ATP hydrolysis [22]. In addition, novel findings on the homodimeric ABC exporter MacB award at least a part of the power stroke for substrate translocation to the ATP hydrolysis step [23].

Based on the variety of TMDs, a new nomenclature was introduced to categorize ABC transporters on the basis of the different TMD folds into seven different types [24]. Type I–III cover classical ABC importers, and type IV to VII mostly cover ABC exporters. A large number of structures and functional information are known for the type IV ABC transporters that are defined by the architecture of Sav1866. Type V also includes a few ABC transporters with importer functions. Type VI ABC transporters are defined by the fold of LptB_2_FG and Type VII by the fold of MacB [24]. These classes of Type VI and VII are of special interest for this review and are complemented by the MlaFEDB system (recently assigned as founding member of type VIII [25]) and the LolCDE system (type VII). Although the classical understanding of the transport cycle from an IF state to an OF state with an occluded state trapping the substrate in a binding pocket is feasible for small substrates such as ions, amino acids, or even smaller peptides, this model cannot apply to large substrates such as lipopolysaccharide (LPS) or lipoproteins. In the following sections, we will summarize the recent findings with respect to the structure and function of these selected ABC transporter systems of MacB, LptB_2_FGC, MlaFEDB, and LolCDE (Figure 3) as a wide range of information is available for these nanomachineries.

## 2. The MacAB-TolC System

The first experimental evidence of an ABC antibiotic efflux transporter in Gram-negative organisms was published in 2001 by Kobayashi et al. [26]. They investigated a system that gave resistance to erythromycin and other macrolide-type antibiotics and was therefore termed macrolide-specific ABC-type efflux carrier, short “MacAB”. Furthermore, they showed that TolC is necessary for MacAB efflux activity [26]. Today it is known that the MacAB-TolC system also exports the extracellular peptide toxin STII [27], the heme precursor protoporphyrin IX [28], cyclic peptides like bacitracin and colistin [29], and penicillin-type antibiotics as well as arsenite [30]. Furthermore, overexpression of MacAB in *Klebsiella pneumoniae* increased resistance against the synthetic tetracycline-class antibiotic eravacycline [31]. All substrates of the MacAB-TolC system are transported from the periplasm across the outer membrane to the extracellular space [27,29]. Interestingly *macB* homologs can be found in Gram-positive bacteria as well, although they lack a periplasm and a second membrane (reviewed in Greene et al. [32]). The expression of *macA* and *macB* is controlled by the PhoP/PhoQ system [33] and therefore downregulated in case of low Mg^2+^ levels, which was shown using real-time quantitative polymerase chain reaction (rt-qPCR) [34].

The structure of the outer membrane protein TolC was determined in 2000 by Koronakis et al. [35]. Due to DNA sequence analysis, it was already proposed in 2001 without having a high-resolution structure that MacB is a transmembrane protein with four transmembrane helices (TMH) and that MacA is a peripheral membrane protein belonging to the membrane fusion protein (MFP) family [26]. Biochemical and biophysical data by Lin et al. revealed in 2009 that MacA stabilizes the tripartite assembly of the MacAB-TolC efflux system through specific interactions with MacB as well as TolC. In addition, it was shown that the N-terminal transmembrane helix of MacA, anchoring it into the inner membrane, is not essential for the functional assembly of the system [36]. In contrast, Tikhonova et al. could show that a MacA mutant lacking the N-terminal transmembrane helix was not able to confer an increase in erythromycin resistance in vivo, although in vitro studies showed that this mutant can still interact with MacB [37]. Furthermore, MacA increases MacB’s affinity for ATP and the substrate erythromycin. Mass spectrometry (MS) and atomic force microscopy (AFM) revealed a dimeric organization of MacB [36]. Bound lipids such as phosphatidylethanolamine (PE) and cardiolipin could only be observed for the dimeric protein using detailed MS analysis [38]. In 2009, the first crystal structure of the periplasmic domain of MacB [39], as well as the crystal structure of a hexameric arrangement of the periplasmic part of MacA from *Actinobacillus actinomycetemcomitans* [40], became available. Further analysis of the interface between MacA and TolC supported a strongly conserved tip-to-tip interaction between those two proteins [41,42,43]. In 2017, the single-particle cryo-EM structure of the fully assembled MacAB-TolC efflux system (see Figure 4) from *Escherichia coli* (*E. coli*) was published in a nucleotide- and substrate-free conformation by Fitzpatrick et al. [44]. This structural information on the MacAB-TolC system was complemented by an ADP-bound crystal structure of MacB from *Acinetobacter baumannii* published by Okada et al. [45] and an ATP-bound crystal structure of MacB from *Aggregatibacter actinomycetemcomitans* published by Crow et al. [29] in the same year (see Figure 5). Moreover, the crystal structure of a Gram-positive MacAB-like efflux pump from *Streptococcus pneumoniae* was published by Yang et al. in 2018 [46].

The structure of the full MacAB-TolC efflux system was derived by Fitzpatrick et al. [44] from a hybrid electron density map combining density maps from two different stabilizing approaches of the tripartite assembly. The first approach was a fusion of the C-terminus of MacB to the N-terminus of MacA as they were expected to be in close proximity. Interestingly, the fusion construct resolved only two copies of MacB, which were forming the dimer in the inner membrane. The other four copies of MacB, each fused to one MacA, were not resolved in the electron density map but produced diffuse density in the two-dimensional classifications. The second approach was a stabilization using disulfide bonds between MacA and MacB introduced via cysteine mutations. Only two of the MacA protomers formed a disulfide bond with MacB. Both approaches stabilized MacAB-TolC sufficiently for the acquisition of high-resolution structural data without loss of functionality in vivo [44]. This confirmed the overall organization of the MacAB-TolC system comprising a dimer of MacB in the inner membrane linked to a hexamer of MacA in the periplasm, which in turn is connected to a trimer of TolC in the outer membrane. This tripartite assembly with approximately 320 Å spans the inner and outer membrane in vivo [44].

Within the fully assembled complex, MacA adopts a hexameric structure, as seen for MacA alone [40], with four domains: a cylindrical α-helical hairpin domain, a ring-forming lipoyl domain, a ring-forming β-barrel domain, and a membrane-proximal domain. A density for the N-terminal transmembrane helix was missing, just as for the already existing crystal structure [40]. The already proposed tip-to-tip interaction between the trimeric TolC and MacA [41,42] was located between the α-helical hairpin region of MacA and the intra- and inter-protomer grooves of TolC [44]. A loop in the lipoyl domain of MacA allows highly conserved glutamine residues [40] to form an inter-protomer hydrogen-bond network that acts like a gating ring. Molecular dynamics simulations suggested that this gating ring acts as a one-way valve for the outward-directed transfer of the substrate [44]. The β-barrel domain and the membrane-proximal domain from three MacA protomers interact with the periplasmic domain of one MacB protomer [44].

MacB forms a homodimer in the tripartite assembly, with each protomer consisting of a NBD, a TMD, and a periplasmic domain (PD). The NBD is connected to the TMD via a long loop and an amphipathic helix. The TMD of MacB is unlike other ABC transporters, built up by only four TMH. TMH1 and TMH2 are elongated and reach above the membrane plane into the periplasm. In between these two helices, the PD is located. This PD contains a so-called porter domain that is a structural homolog to the AcrB “porter domain” and a sabre (small alpha beta-rich extracytoplasmic) domain. The porter domain is formed by two subunits that are located before and after the sabre domain. TMH3 and TMH4 are shorter than TMH1 and TMH2 and are connected via the so-called shoulder loop [29]. The major coupling helix is located in between TMH2 and TMH3; the minor coupling helix is located C-terminal of TMH4 [44]. The major and minor coupling helix interact with the NBD from the same protomer as there is no domain swapping [29]. The role of the minor coupling helix is not fully understood as, according to Crow et al. [29], the deletion of this helix did not influence MacB activity significantly, but for Okada et al. [45], deletion of the minor coupling helix resulted in the loss of drug export.

Although substrate-bound structures are lacking, the nucleotide-free MacAB-TolC cryo-EM structure [44], together with the ATP-bound crystal structure [29] and the ADP-bound crystal structure [45], allow prediction about the function of MacAB-TolC mediated efflux.

In the nucleotide-free state (see Figure 5A, left panel), the transmembrane dimer interface adopts a V-shape form having the periplasmic parts of TMH1 and TMH2 of one protomer far away from the other protomer and thereby forming a cavity on the periplasmic site margined by the periplasmic domains of MacB. This arrangement also brings the periplasmic domains into an open conformation forming a small gap towards the periplasm (see Figure 5B, left panel). In the cryo-EM structure, an additional electron density was observed in this periplasmic cavity between the periplasmic elongations of the TMHs. The orientation of this additional density already indicates a lateral substrate entrance. Nevertheless, it was not possible to examine the identity of this molecule. The NBDs are also separated and far away from each other in the nucleotide-free structure [44]. In the ATP-bound state (see Figure 5A, middle panel), the NBDs are dimerized and form a tightly packed classical head-to-tail arrangement. TMH1 and TMH2 of each protomer align parallel to each other in a rigid dimer interface, omitting the V-shaped form but adopting a so-called “zipped stalk” conformation. The PD adopts a closed form without an opening towards the periplasmic side (see Figure 5B, middle panel). The NBDs dimerization is supposed to mediate the “zipping” of the stalk and closure of the PD via the major coupling helix and movement of TMH2, which changes the dimer interface favoring close proximity of TMH1 and TMH2 of each protomer. This transfer of structural movements from one side of the membrane to the other via the transmembrane helices is referred to as mechanotransmission. The mechanotransmission mechanism was investigated in vivo by using cysteine mutants that lock MacB TMH1 and TMH2 in the zipped conformation. Those mutants showed decreased resistance to erythromycin [29]. In the ADP-bound state (see Figure 5A, right panel), the NBDs are dimerized but not as tight as in the ATP-bound state. TMH1 and TMH2 adopt a V-shape-like open conformation as in the nucleotide-free state, but the PDs arrange differently, and the opening towards the periplasm is absent (see Figure 5B, right panel) [45].

Biochemical data of Tikhonova et al. showed that MacB exhibits basal ATPase activity in detergent, which is little to none affected by the addition of either substrates or MacA or TolC [37]. When reconstituted into proteoliposomes, the basal ATPase activity of MacB is reduced about 10-fold but is strongly stimulated in the presence of MacA. MacA mutants missing either an N- or C-terminal part of MacA could not stimulate MacB ATPase activity in proteoliposomes in vitro and were also not able to increase erythromycin resistance in vivo, although the N-terminal truncated mutant was capable of binding MacB in vitro [37]. Modali et al. revealed that MacA stabilizes the ATP-bound state of MacB, and a single mutation in the membrane-proximal domain of MacA abolishes the macrolide efflux function of MacAB-TolC [47]. Lu et al. discovered that the periplasmic domain of MacB is essential for MacA-dependent stimulation of MacB ATPase activity. Furthermore, ATP binding of MacB increases affinity towards MacA [48]. In 2021, Souabni et al. determined a transport rate of three molecules per hydrolyzed ATP molecule for the substrate roxithromycin [23]. Recently, Batista dos Santos et al. [49] showed that the presence of TolC increases MacB ATPase activity in detergent as well as in lipid nanodiscs. Furthermore, they could also show a substrate-induced increase in MacB ATPase activity for the fully assembled MacAB-TolC system in a lipid environment [49].

Based on the available structures, Crow et al. [29] postulated a mechanism for substrate efflux of the MacAB-TolC system in 2017 termed the “molecular bellows” mechanism. In the nucleotide-free state, the cavity between the periplasmic part of TMH1 and TMH2 of MacB and MacA is open to the membrane, and substrates can enter. Upon ATP binding, the dimerization of the NBDs on the cytosolic side causes closure of the TMDs of MacB and constrains the volume of the cavity. This generates pressure and pushes the substrate through MacA towards TolC. The gating ring inside MacA prevents backflow of the substrate once the pressure is balanced. After ATP hydrolysis, the system switches back to the nucleotide-free open state [29]. This mechanism was further expanded by Souabni et al. [23] in 2021, investigating the role of ATP hydrolysis using a quantum dots-based real-time analysis of substrate transport and ATP turnover. They suggested a “modified bellows mechanism” where ATP hydrolysis is additionally energizing the transport of the substrate across the MacA gating ring towards TolC. In this modified mechanism, MacA and its transmembrane helix act as a mediator for the additional energetic input. In their study, substrate translocation and ATP hydrolysis are shown to be synchronous events [23].

Although much is known due to biochemical and structural data, many open questions remain regarding the function of the MacAB-TolC system. Examples are: How is the substrate recognized? Is there a specific substrate binding site? Is there a feedback mechanism from the periplasm to the NBDs? How is lateral substrate leakage prevented? These questions require further research using intermediate or substrate-bound structures.

## 3. The Lpt System

The cell envelope of Gram-negative bacteria exhibits a complex architecture. It not only consists of an inner membrane (IM), made up of phospholipids in the inner and outer leaflet, and an additional outer membrane (OM) but also a periplasm in between with a cell wall formed by crosslinked peptidoglycan. The OM protects the cell from the environment and serves as a potent barrier for hydrophobic molecules. Its composition is highly asymmetrical, with glycerophospholipids in the inner leaflet and mainly LPS molecules in the outer leaflet [50,51]. The interested reader is referred to comprehensive reviews focusing on the structure and synthesis of LPS [52,53,54], as it will be only briefly described here. The LPS structure can vary among different bacteria; typically, it can be divided into three parts: LPS is anchored to the membrane via its Lipid A moiety, a β-1′-6-linked glucosamine disaccharide, every sugar being acylated with fatty acid chains. Additionally, in *E. coli*, the glucosamine is phosphorylated at positions 1 and 4′ and can be further modified, e.g., upon polymyxin exposure with ethanolamine or 4-amino-4-deoxy-L-arabinose to decrease the negative net charge [55,56]. The core oligosaccharide moiety is connected to Lipid A via 3-deoxy-D-manno-oct-2-ulosonic acid (Kdo), which itself is linked with several heptose and hexose molecules. On top of the core oligosaccharide is the O antigen, also called O-antigenic polysaccharide (O-PS); it represents the terminal part of LPS and, as the name suggests, is a polymer made up of different kinds of oligosaccharides. The composition of the O antigen varies greatly, not only between different but also within the same species. *E. coli*, as one example, displays over one hundred different serotypes [52]. LPS is found in most, but not all, Gram-negative bacteria, with exceptions like *Sphingomonas paucimobilis* and *Treponema pallidum* [57,58]. In *E. coli* and *Salmonella*, the presence of LPS is essential, although this is not the case for all LPS-containing bacteria. Certain strains of *Neisseria*, *Moraxella,* and *Acinetobacter* can live without genes necessary for the synthesis or transport of LPS to the OM [59].

The LPS homeostasis displays a delicate challenge. Even though it is exclusively present in the outer leaflet of the OM, the biogenesis starts on the complete opposite side of the cell envelope: The inner leaflet of the IM (and the cytoplasm). Therefore, it does not only need to be extracted from the IM—an energy-consuming process—but the amphipathic LPS molecule also needs to traverse the aqueous periplasm on its route to the OM. Ultimately, the LPS needs to be incorporated into the outer leaflet of the OM. This process must occur repetitively during the life cycle of the cell in a highly ordered manner, as processes in the periplasm and outer membrane cannot be energized directly through ATP hydrolysis.

This part of the review will deal with the LPS transport (Lpt) machinery (see Figure 6), which mediates the extraction of LPS from the IM, its transport across the periplasm, and insertion into the OM. A special focus lies on the ABC transporter, including a step-by-step examination of its functionality and recent discoveries.

Intensive research on the LPS transport over more than five decades, starting with genetic approaches along with biochemical and structural studies, gradually revealed the Lpt system to be a multiprotein assembly with an ABC transporter LptB_2_FGC in the inner membrane [60] and a translocon LptDE in the outer membrane [61] which are connected by a periplasmic bridge built up by (most likely) oligomeric LptA [62] (see Figure 6). In this complex, LptB_2_FGC provides the energy for the extraction and transport of LPS by ATP binding and hydrolysis. Although its TMD fold is reminiscent to type V ABC transporters, LptB_2_GFC is considered to be the first member of the new type VI ABC transporter family, as there are several aspects that led researchers to classify this transporter into its distinct group [24,63].

In the domain organization of the LPS extractor, the two homologous proteins, LptF and LptG, form the TMD [64,65]. Both proteins have an additional periplasmic domain, the so-called β-jellyroll domain. This protein fold is a hallmark of the Lpt system, as it occurs in five of the seven Lpt proteins (LptA, LptC, LptD, LptF, and LptG; see Figure 6). The homo-dimeric LptB shows the fold of a canonical NBD and serves as the motor domain of the extractor. Its structure as an isolated protein was resolved by two independent groups in the same year, Sherman et al. and Wang et al. [66,67]. It is present as a dimer in the cytoplasm, where it binds and hydrolyzes ATP to energize the LPS translocation and forms the functional transporter with LptFG [60]. What makes the LPS transporter complex unique is the presence of a mysterious protein within the inner membrane complex: LptC. It is a small protein, only consisting of an N-terminal TMH and a periplasmic β-jellyroll domain [68]. Like all other Lpt proteins, it is essential for LPS transport, as deletion strains show phenotypes attributed to defective LPS transport, such as increased sensitivity to hydrophobic compounds [69].

LptD and LptE build the translocon in the outer membrane [61,70,71,72,73]. LptD is a large, 26-stranded β-barrel protein in which the lipoprotein LptE resides, forming a barrel-and-plug complex. The N-terminal part of LptD features a β-jellyroll domain as well and is connected to the jellyrolls of LptB_2_FGC via the periplasmic LptA, in which this protein fold was first discovered [74]. The N-terminal part of LptAs β-jellyroll domain interacts with LptC, while its C-terminal part showed crosslinks to the β-jellyroll domain of LptD. Studies also showed that LptA is capable of forming oligomeric structures in vivo and in vitro, although the exact number of LptA molecules necessary for establishing the periplasmic bridge between the complexes in the IM and OM remains unknown [74,75,76].

Early models based on initial structures of LptB_2_FG by Dong et al. and Luo et al., who only had the transporter in a single functional state available, proposed a sequence in which ATP binding would open the cavity, allowing LPS to enter and subsequent ATP hydrolysis and release of ADP would push LPS out of the transporter [65,77]. Today, additional structures of LptB_2_FG together with LptC, bound LPS, and/or bound nucleotide by the groups of Li, Luo, Tang, and Owens suggest the following model [78,79,80,81]: In the initial state of the transport cycle, the TMH of LptC (LptC^TM^) resides between LptF and LptG (see Figure 7, state i). LPS enters the cavity first (see Figure 7, state ii) and triggers in a not completely understood manner the release of LptC^TM^. Here, a strict NBD-TMD coupling is apparently involved (see Figure 7, state iii). By this, the cavity narrows, and its residues bind tightly to the LPS molecule and elevate it inside the cavity. The NBDs of LptB_2_ bind tightly to each other, which results in a complete collapse of the cavity and expulsion of LPS (see Figure 7, state iv). ATP, which can bind before ejection of LptC^TM^, as new data suggest, is hydrolyzed to ADP and released from the NBDs. This reopens the cavity, and LptC^TM^ can bind again between TMH5 of LptF (THM5_F_) and TMH1 of LptG (TMH1_G_) to allow the next transport cycle (see Figure 7, state v) [82].

Together, LptF and LptG are adopting a V-shaped fold with an opening to the periplasm and only a few contacts between their interfaces, namely TMH1 and TMH5 (see Figure 7B and C). The limited interaction sites suggested early on that one, or both interfaces may open further to allow the lateral entry of LPS into the cavity [65,77]. The formed cavity is covered with residues of hydrophobic amino acids, which was confirmed by the structures of LptB_2_FG(C) with bound LPS. Additional charged residues stabilize bound LPS via salt bridges to the phosphate groups and glucosamine disaccharides of the lipid A moiety as well as the core oligosaccharide [78,79]. Even before the structure of LptB_2_FGC with LPS was revealed, Hamad et al. and Bertano et al. used mutational studies and bacterial strains, which constitutively modify lipid A phosphates to show that a cluster of charged residues in the TMH1 of LptG is important for LPS transport, most likely by binding LPS through establishing contact sites with the phosphate moieties of lipid A [83,84]. The first step in the LPS transport is the entry of the LPS molecule into this cavity for extraction, and this process already raises two questions: (i) From which side of the transporter does LPS enter? The structure of LptB_2_FGC allows LPS entry in principle from both LptFG interfaces, TMH1_F_:TMH5_G_ or TMH5_F_:TMH1_G_. (ii) How does the transporter differentiate between LPS and other phospholipids, which are also present in the outer leaflet of the inner membrane?

The first question was answered by crosslinking studies using unnatural amino acids as photo-crosslinkers, together with structures of LptB_2_FGC, in which the periplasmic domains of LptF and LptG were resolved. Owens et al. detected crosslinks of LPS to residues in TMH1 of LptG and TMH5 of LptF as well as LptC, but not with the possible cavity opening formed by TMH1 of LptF and TMH5 of LptG [81]. Moreover, structures of the LPS ABC transporter with LptC revealed that the LptC^TM^ is positioned between the interface of TMH5_F_:TMH1_G_, which is in line with the aforementioned crosslinking studies. So far, no data supported the proposal that LPS might enter from the TMH5_G_:TMH1_F_ interface or LptC to reside in that position. Additionally, the structure of LptB_2_FGC showed that the β-jellyroll domains of the transporter are placed above the interface formed by TMH1_F_ and TMH5_G_, possibly blocking the entry of LPS from this side due to the bulky core oligosaccharide and O-antigen moieties [81].

The second question, why phospholipids are not transported by the Lpt system, is not easily answered. The aforementioned study by Owens et al. showed that even in the absence of ATP, LPS is able to enter the cavity of the transporter [81]. One possibility is that phospholipids can enter the cavity through the TMD interface as well but are not recognized by the transporter as a substrate due to the lack of interactions mentioned above. Since the simplest LPS structure, enabling cell viability, contains lipid A and Kdo, it is likely, that the presence of these key components is necessary for substrate recognition by the Lpt transporter [52,85].

We know now that the TMH5_F_:TMH1_G_ interface—in which LptC^TM^, at least at some point of the transport cycle, resides—is the entry for LPS. However, how does LPS enter the cavity of LptFG with a TMH of another protein in its way? The LptB_2_FG(C) structures of Li et al. revealed that LptC^TM^ pushes away TMH1-3 of LptG, preventing the interaction of bound LPS with the charged residues mentioned above (see Figure 7B and C). The tighter binding of LPS might therefore push LptC^TM^ out of the cavity [79]. The constriction of the cavity ultimately results in its full collapse and the transfer of LPS from the cavity to the β-jellyroll domains. This movement, as well as the reopening of the cavity, is performed by the NBDs LptB_2_, which bind and hydrolyze ATP. Closure and opening of the NBD interface are thought to be tied to the collapse and opening of the TMD cavity by a rigid body mechanism, as suggested for other transporters as well [5,86,87]. The LptB structure features a groove region in which the coupling helices of LptF and LptG are embedded [66]. These coupling helices are conserved in the TMD of ABC transporters and are crucial for the TMD-NBD interaction and can be found between TMH2 and TMH3 in LptFG [7,65,77]. They were identified using photo-crosslinking and mutational studies, in which defects induced by substituting a conserved glutamate in both LptF (E84) and LptG (E88) could be suppressed by altering an amino acid in the groove region of LptB (R91) [88]. Interestingly, the same study showed that LptF and LptG do not act symmetrically, as identical changes at equivalent positions led to different defects. Following the signature motif is the signature helix, in which Simpson et al. identified an arginine residue (R144) to be important for forming contacts with the Q-Loop and Walker B motif. Altering this residue led to reduced LPS extraction due to a lower ATP binding affinity, favoring the open conformation of the transporter. Strikingly, an alteration in the C-terminal domain (CTD) of LptB (F239), a domain unique to this transporter, is usually lethal but could complement the change in the signature helix. The data for this mutation showed a decreased ATP hydrolysis, indicating that this mutant is favoring the closed conformation, and the combination of these two defects complement each other, resulting in a functional transporter with decreased ATPase activity. Since full restoration of ATP hydrolysis in this double mutant is not necessary for LPS transport, the researchers proposed that the binding of ATP leads to the collapse of the LptFG cavity and hydrolysis to its reopening [89]. This correlation is supported by structures of the transporter in its closed form with bound β-γ-imidoadenosine 5′-triphosphate (AMP-PNP), a nonhydrolyzable ATP analog, and with bound ADP-vanadate [78,79]. ATP binding leads to the closure of the LptB_2_ dimer interface and the concomitant anti-clockwise rotation of the LptFG TMH1-5, ultimately closing its cavity [78].

Since this review focuses on the transporter of the Lpt system, subsequent transport steps will be only briefly described here. At the end of each transport cycle, a new LPS molecule is placed into the β-jellyroll domains of the transporter. The functionality of the transport system is often compared with a PEZ candy dispenser, where the LptB_2_FGC transporter acts like a spring, with each cycle loading a new LPS “candy” onto the bridge, pushing the former one further ahead on the LptA bridge towards LptDE and the OM [90]. Although both LptF and LptG feature β-jellyroll domains, structural, mutational, and crosslinking studies so far only showed that LPS travels from the cavity to the β-jellyroll domain of LptF, but not LptG, and further to the one of LptC [81]. The backflow into the cavity must be prevented if the entry of new LPS is not much faster since LPS does not diffuse away from the transporter. Owens et al. proposed on the basis of their LptB_2_FGC structure that the β-jellyroll domain of LptF can adopt a closed conformation, preventing a backward flow much like a valve [81]. Another possibility for the unidirectional flow might be different binding affinities of LPS for the different proteins, as this was at least shown for the transfer of LPS from LptC to LptA [68]. Even though LptA is known to form oligomers in vivo and in vitro, the exact number forming the bridge is unknown [62,74]. Sherman et al. succeeded with the in vitro reconstitution of the complete system, proving that LptA physically connects LptB_2_FGC with LptDE [91]. Once LPS reaches LptDE in the outer membrane, it is first placed into the β-jellyroll domain of LptD. The lipid A moiety of LPS is proposed to enter the membrane directly through a cavity between its β-jellyroll domain and the 26-strand β-barrel. The hydrophilic part of LPS would then first enter the β-barrel from the periplasmic side before exiting it through a lateral gate between helices 1 and 26, which showed to have only a few interacting residues [73,92,93]. LptE, the barrel’s plug, is not only important for the biogenesis and proper folding of LptD but was also shown to bind LPS and extract it from aggregates, suggesting that its role is to accept LPS coming from the periplasm, weaken neighboring LPS-LPS interactions and assisting its insertion into the OM [61,93,94,95,96,97]. Once the hydrophilic part of LPS has passed the lateral gate, it becomes part of the already existing LPS network. Intriguingly, there is evidence that the activity of the LPS transporter LptB_2_FGC in the inner membrane is influenced by the translocon LptDE. For an in vitro setup, inhibition of LPS transport and ATPase activity of LptB_2_FGC in liposomes was observed when adding LPS-preloaded LptDE-containing liposomes [98]. Lately, LptB_2_FGC has been shown to exhibit an adenylate kinase activity in addition to the ATPase activity, as it was reported for other ABC transporters like MsbA and CFTR [99,100,101].

Due to recent advances in structural biology, many answers regarding LPS transport could be answered. However, still some key questions remain, especially regarding the enigmatic LptC: What role does LptC play in the transport process since no other ABC transporter features a protein alike? Even though LptC is found in structures with LptB_2_FG, its TMH is neither required for LPS to enter the cavity nor for transport in general. Studies showed that LptC variants, in which the TMH was deleted, are still able to form a complex with the other Lpt proteins to transport LPS [102]. Experimental data suggest that the LptC^TM^ plays a regulatory role in the ATPase activity of the transporter. In vitro, LptB_2_FG displays an increased ATP hydrolysis when no LptC or only variants without TM helix (LptC^∆TM^) are present. Therefore, the helix might reduce futile ATP hydrolysis by coupling it efficiently to substrate extraction [78,79,81]. Lethal LptC deletions can be suppressed when the arginine residue R212 in the β-jellyroll domain of LptF is substituted with glycine, restoring wildtype LPS transport and ATPase activity of the transporter, even though affinity to LptA is reduced [103,104]. Interestingly, this point mutation is still able to form a complex with LptC when it is present. This indicates that the periplasmic β-jellyroll domain of LptC is responsible for the interaction with the other Lpt components and enhances the stability of the transporter to the periplasmic bridge. The availability of LPS-bound LptB_2_FG structures with and without LptC allows a comparison of the cavity and interactions with the LPS molecule. With the LptC^TM^ missing, the transporter cavity is significantly smaller (see Figure 7C). Simultaneously, the LPS structure in those structures is better resolved, indicating an improved binding of LPS once the LptC^TM^ is removed from the transporter. When, how, and why the LptC^TM^ is removed during the transport cycle is still not completely clear.

Recently, Wilson et al. approached these questions by combining LptC^∆TM^ with mutations in other Lpt transporter components and searched for synergistic or suppressive effects on the phenotypes. They aimed to elucidate the exact role of LptC^TM^ in the different steps of the extraction cycle, as deletion of the TM helix should strengthen or weaken the phenotype when a mutant is affected in the same step, while no change to the phenotype was expected when LptC^∆TM^ was combined with mutants, in which the TM helix plays no role. Two observations were made: (i) The presence of the LptC^TM^ increases the stability of the protein, possibly by benefiting from a complex formation with LptB_2_FG, and ii) the phenotype of LptB and LptF/G mutants with defects in ATP binding and NBD-TMD coupling were affected, indicating that the LptC^TM^ plays a role in these steps. This new data hints that i) binding of ATP to LptB can occur with the LptC^TM^ associated with LptFG (although it is not obligatory) and (ii) the coupling helices in LptFG, as well as the corresponding groove region in LptB, take part in LptC^TM^ displacement from the transporters cavity, meaning that the sole entry of LPS into the cavity is not sufficient for this process [82]. The proposal that ATP can bind to LptB_2_FG with LptC^TM^ still present is contrary to former models, in which LPS binding and formation of tighter contacts to LptFG is displacing LptC^TM^ with ATP binding afterwards, particularly as structures of LptB_2_FG with bound LPS and displaced LptC^TM^ were lacking nucleotides so far [79]. It is surprising that the essential process of LPS extraction, which takes place countless times during the lifetime of a cell, features an apparently useless step. One possible explanation is that the intercalation of LptC^TM^ slows down ATP hydrolysis [78,79,81,87,99] to synchronize the hydrolysis of ATP with the binding of LPS to the cavity of the extractor, preventing futile ATP hydrolysis. It also remains enigmatic if the placement of LptC^TM^ is a step that is not performed during active LPS transport but rather to slow it down before turning it off.

## 4. The Mla System

As mentioned in the previous section, the OM is an asymmetric bilayer, and all the components for building the OM envelope are synthesized either in the cytoplasm or the IM before being transported across the periplasm to be inserted into the OM [105]. As mentioned in the previous section, LPS is trafficked across the cell envelope via the Lpt system. The *m*aintenance of *l*ipid *a*symmetry (Mla) pathway is involved in maintaining the asymmetry of the OM by trafficking phospholipids (PLs) between the IM and OM [106,107,108]. The Mla pathway is a multi-component system and uses a ferry-like mechanism to shuttle phospholipids across the periplasm. Its mutation leads to the accumulation of PLs in the outer leaflet of the OM, increased OM permeability, and increased susceptibility to antibiotics [106,107,108,109]. Although translocation of phospholipids between outer and inner membrane was initially discovered in *Salmonella typhimurium* as early as 1977, it was not until 2009 that the components of the phospholipid transport system in Gram-negative bacteria were discovered via homology of conserved transporters in Actinobacteria and chloroplasts [106,110,111,112]. The orthologous TGD pathway of plants transports phosphatidic acid from the OM to the IM of chloroplasts [110], while the Mce4 pathway in Actinobacteria is paralogous and imports exogenous cholesterol [111,112].

In *E. coli*, the genes representing the Mla system are located on the *mlaFEDCB* operon, which is conserved among Gram-negative bacteria. Although *mlaA* and *ompC/G* are part of the Mla system, these genes are located outside the *mla* locus [106].

The Mla system is a six-component system with components present in each compartment of the cell envelope. It includes an OM lipoprotein termed MlaA; the OM major porins OmpF/OmpC, which act as a scaffold for MlaA; a soluble periplasmic component known as MlaC, which acts as a carrier of PLs, shuttling them between the membranes; a mammalian cell entry (MCE) domain protein called MlaD which is anchored in the plasma membrane; the transmembrane domain of the ABC transporter MlaE; a sulphate transporter and anti-sigma factor antagonist (STAS) domain protein called MlaB and the ATPase MlaF. These components create the following three main parts of the Mla system [113,114]: (a) the trimeric porin OmpC, which forms a complex with lipoprotein MlaA at the OM; (b) a soluble lipid-binding protein, MlaC, located in the periplasm; and (c) MlaFEDB, an ABC transporter localized in the plasma membrane (see Figure 8).

MlaA is a lipid transport protein and assembles into a ring-shaped α-helical structure that contains a central pore [115]. It forms a complex with both the OM porin proteins, OmpC and OmpF [115,116]. However, the complex of OmpC-MlaA is the active species, as MlaA copurifies with OmpC. MlaA binds in the groove between the two OmpF/C monomers, and the interaction between MlaA and OmpF/C is mainly mediated via van der Waals forces [115]. Since MlaA might be unstable on its own in the lipid bilayer, the porins might function as a scaffold to ensure the proper functioning of MlaA [115].

MlaC is a periplasmic lipid-binding protein. The crystal structure of MlaC was resolved and comprised four ß-sheets and seven helices with a large hydrophobic pocket in the core of the protein. Since MlaC can bind to both IM and OM complexes, it probably exhibits a central role in the transport of PLs between the membranes [117]. MlaC has a high affinity towards phospholipids and can bind three different PLs: phosphatidylglycerol, phosphatidylethanolamine, and cardiolipin [118,119].

MlaD is anchored to the IM through a single N-terminal TMH with its MCE domain residing in the periplasm. The MCE domains have been involved in lipid uptake in Gram-negative bacteria and retrograde transport of PLs in chloroplasts [110,117,120]. MlaD forms a ring-shaped homo-hexamer with a central hydrophobic pore that allows movement of PLs [114,117,118,119].

MlaE represents the transmembrane domain of the ABC transporter, which forms a homodimer. In contrast to the previously described TMDs of MacB and LptF/G (Section 3 and Section 4), MlaE has five TMHs. Recently, it has been assigned as a founding member of the type VIII group of ABC transporters [24,117]. Each subunit contains one elbow helix (EH), five transmembrane helices (TMH1–5), one coupling helix (CH), and one periplasmic helix (PH). TMH1, TMH2 and TMH5 of the two subunits of MlaE form a central hydrophobic cavity. EH runs parallel to the inner membrane plane while the CH connects TMH2 and TMH3 and is involved in interaction with MlaF. The PH is placed between TMH3 and TMH4 and interacts with MlaD [121].

MlaF represents the nucleotide-binding domain of the ABC transporter with a conserved structure representative of the ABC superfamily and is present as a dimer [121]. MlaB contains a STAS domain and represents the accessory protein, which is involved in cross-talk with the NBDs of ABC transporters. The two MlaB subunits are located on the opposite side of the MlaF dimer [122,123] and are involved in stabilizing the complex and ATP hydrolysis [109,124].

The inner membrane complex (IMC) of the Mla pathway is composed of MlaFEDB, which represents the ABC transporter. Overall, the complex comprises of four different proteins: MlaF, MlaE, MlaD, and MlaB in a stoichiometry of 2:2:6:2 [109,117,121,125]. The homodimers of MlaE and MlaF function as the TMDs and the NBDs, respectively. Associated with the homodimers are the unique auxiliary proteins: MlaB and MlaD. While there are two copies of MlaB, MlaD is present as a hexamer in the complex [109]. The two MlaB proteins in the cytoplasm do not contact each other, and each MlaB molecule interacts with one MlaF [121]. The MlaD hexamer rests on top of the periplasmic side of the MlaE dimer [117]. The six α-helices form a hollow hydrophobic channel to allow the transport of lipids. A total of six MlaD TMHs in the MlaFEDB complex are inserted into the membrane [121]. Each three MlaD subunits incorporate one MlaE where the interaction between the TMH of MlaD and the EH of MlaE is critical for the phospholipid transport function of MlaFEDB [121].

Recent cryo-EM structural work on the Mla system has provided further insights into this intriguing complex. The high-resolution structures from *E. coli* [114,121,126], *Pseudomonas aeruginosa* [127], and *Acinetobacter baumannii* [128,129] clearly indicate that the overall architecture of the whole complex is conserved throughout these species.

These structures show that in the absence of ATP, MlaE embraces a V-shaped open conformation to frame a cavity with the wider side confronting the hydrophobic channel of MlaD. In the case of *E. coli*, the lipid binding site has been shown to be the outward-open pocket of MlaE [114,126], while for *A. baumannii*, lipid binding is described to happen at the pore of MlaD and in between the pore loops of the MCE domain of the MlaD hexamer [128]. A few of the structures contain residual electron density in the cavity and in different locations and could represent PLs or bound detergent [114,121,126,127,128,129] (see Figure 9A). The direction of lipid transport by the Mla system is discussed as controversial. A retrograde transport mechanism was originally described for the Mla system, where it maintained the lipid asymmetry by removing mislocated phospholipids from the outer leaflet of the OM and importing them back to the plasma membrane [106,107,108,109,113,115,116,130]. However, there is also data that supports the export of phospholipids (anterograde transport) from the IM to the OM [119,125,126,131] or even a bi-directional transport between both membranes [114].

For simplicity, we only describe here the proposed retrograde transport mechanism in the context of phospholipid import (see Figure 9B). However, reversing the outlined steps could result in anterograde transport of phospholipids, as also described [119,125].

As part of the OmpF/C complex, MlaA is embedded inside the OM, and together they form a channel across the membrane [115]. Phospholipids are extracted from the outer leaflet of the OM into the channel via a lateral pathway. Interaction of MlaC to the MlaA-OmpC/OmpF complex results in phospholipid transfer to the hydrophobic pocket of MlaC [114,117,118]. Then, MlaC diffuses across the periplasm to deliver the lipids to the MlaFEDB complex in the IM. A hexameric ring formed by MlaD subunits creates a central hydrophobic tunnel for the transport of lipids [117]. MlaC acts as a chaperone and directly binds to MlaD [117], transferring the lipid into a continuous channel from MlaD to an outward-facing MlaE [114,121,127,128,129]. ATP binding induces a conformational change in MlaE, resulting in the collapse of the lipid-binding pocket, thereby facilitating the incorporation of the lipids into the IM [121]. The auxiliary protein MlaB is known to regulate the transport [109,124].

Although the recent advances in research have added a plethora of information about the Mla system, a major unresolved question is still the directionality of lipid transport. MCEs are usually involved in the retrograde transport of misplaced phospholipids; however, there are still crucial details missing regarding the transfer of PLs from MlaC to MlaD.

Another open question is the reason behind the formation of stable complexes of MlaA with both OmpF and OmpC. Interestingly only one of these assemblies is functional, although functionality towards the transport of PLs lies within MlaA. These questions need to be addressed in the future to better understand the system.

## 5. The Lol System

Lipoproteins are crucial elements in bacteria. They are either located in the outer leaflet of the cytoplasmic membrane or in the leaflets of the OM. The latter case is most often true for Gram-negative bacteria, which is also the focus of this section. Lipoproteins are a compelling object of study for many reasons. One of them is their strong involvement in building and maintaining the OM of Gram-negative bacteria. As the OM is the first line of defense against xenobiotics, lipoproteins, their synthesis, and their transport pathway are attractive targets for novel antibiotics [132,133]. This section focuses specifically on the ABC transporter, which is involved in the transport of lipoproteins, and shows a transport mechanism differing from the classical ABC transporter mechanism. For a better overview regarding the synthesis, sorting, and function of lipoproteins, the interested reader is referred to other reviews [132,133], as these aspects will only be briefly mentioned here.

Lipoproteins are synthesized in the cytoplasm together with a cleavable signal peptide, which contains a consensus sequence that is highly conserved among lipoproteins, called the lipobox. The consensus sequence is L-A/S-G/A-C and was confirmed by various sequence analyses [134,135]. After translocation to the IM via the Sec [132,136] or, in some cases, via the Tat machinery [132,137], lipoproteins undergo further maturation steps in the outer leaflet of the IM. In the first step, the enzyme Lgt attaches a diacyl moiety to the cysteine in the aforementioned consensus sequence [138]. Subsequently, the enzyme Lsp [139,140] cleaves the signal sequence from the cysteine, and therefore this residue is named Cys^+1^ and becomes the new N-Terminus of the lipoprotein. In the last maturation step, another acyl group is attached to the free amino group of the cysteine by Lnt [141,142]. The sorting of lipoproteins either to the IM or the OM is determined in *E. coli* by the amino acid next to the N-terminal cysteine of mature lipoproteins (the +2 position). If this amino acid is aspartate, the lipoprotein is retained in the IM [143]. In the case of most other naturally occurring residues, the lipoprotein is localized to the OM [144].

The trafficking of lipoproteins to the OM of Gram-negative bacteria is mediated by the ***l***ocalization *o*f *l*ipoproteins (Lol) system [133]. The discovery of the Lol system started with the discovery of the periplasmic chaperone LolA (called p20 at that time), which was found to form a soluble complex with a lipoprotein [145]. Soon, the interaction partner of LolA, namely LolB, was also discovered [146]. LolB was localized to the OM and was identified as an essential protein for *E. coli*, as depletion of LolB is lethal. It was further shown that the incubation of a lipoprotein-LolA complex with a soluble LolB derivative led to the transfer of the lipoprotein to the OM. Through these findings, an initial mechanism for lipoprotein localization was already emerging. Soon after that, the crystal structures of LolA and LolB were solved [147]. Remarkably, both proteins showed a similar overall structure, despite their different amino acid sequences. Both proteins show a beta-barrel structure with a hydrophobic inside and alpha-helical lid. The hydrophobic cavities were identified as the possible binding sites for the acyl chains of the transported lipoproteins. This was further supported by finding polyethylene glycol 2000 monomethyl ether (PEGMME2000), which was used for the crystallization, in the hydrophobic cavity in one of the LolB structures [147].

It was later discovered that the detachment of lipoproteins from the membrane is ATP-dependent, and the corresponding protein is an ABC transporter [148]. Yakushi et al. solved the stoichiometry of this transporter termed LolCDE. It is a tetrameric and asymmetric ABC transporter composed of the proteins LolC and LolE, which make up the TMDs, and two copies of LolD, which form the NBDs. It was predicted that both proteins, LolC and LolE, each possess four transmembrane helices and a large periplasmic domain [148]. Due to the topology of LolCDE, the transporter can be assigned to the group of type VII ABC transporters [24]. Taking together all main findings, a general transport cycle, as shown in Figure 10, was derived.

One major unknown factor was certainly the ABC transporter LolCDE. Its mechanism has to surely differ from the mechanism of classical ABC transporter, as the substrate is located in the outer leaflet of the inner membrane and not in the cytoplasm. However, for a relatively long time after the discovery of the Lol pathway, the ABC transporter LolCDE lacked structural analysis, and thus, a detailed understanding of its molecular mechanisms was missing. The first published structure related to LolCDE comprised a crystal structure of a soluble periplasmic domain of LolC solved by the Koronakis lab [29]. This structural analysis was done to confirm the structural similarity of the periplasmic domain of LolC to that of the homologous ABC transporter MacB, which is involved inter alia in the efflux of antibiotic macrolides (described in Section 2). The confirmation of the similarity of the two periplasmic domains of LolC and MacB led to the conclusion that LolC could also follow the mechanotransmission mechanism. This characteristic periplasmic domain is divided into two subdomains named sabre and porter. The superposition of the sabre domains of LolC and MacB showed a prominent loop in LolC which is not present in MacB. Accordingly, this loop was investigated further by the Koronakis lab, and they solved the crystal structure of the periplasmic domain of LolC in complex with the periplasmic chaperone LolA [149]. This structure sheds light on the molecular details of the interactions within the complex and highlights the importance of this loop, which was termed “hook”. Furthermore, an additional important interacting domain was found in LolC, which was termed “pad”. The hook and the pad were determined to be essential for the recruitment of LolA. These findings further underpin the assignment of LolCDE into the group of type VII ABC transporters, as MacB is the founding member of this group [24]. In a more recent study, the crystal structure of LolA bound to a ligand was also solved by the Koronakis lab [150]. The structure shows the precise interaction of the acyl chains of the lipoprotein with the hydrophobic cavity in LolA. In addition, an overlap between the acyl binding sites and the LolC binding sites of LolA were found by comparison with other LolC-LolA structures. This indicates that the substrate binding to LolA is inducing the detachment from LolC. These findings further complete our understanding of the Lol system.

Since 2021, major breakthroughs have been made regarding the structural characterization of LolCDE. Tang et al. started this by solving six structures of the transporter representing different states of the transport via cryo-EM [151]. The structures represent LolCDE solubilized in the detergent LMNG. In the apo state, the NBDs show a relaxed conformation, whereas the rest of the transporter shows a rather compact conformation. More precisely, the transmembrane (TM) segments of LolC twist around the TM segments of LolE, and the PD of LolC rotates to the front of the PD of LolE. In the substrate-bound conformation, the TMDs of LolCDE are outward-opened, leading to a V-shaped central channel. In addition, two lateral gates are observable, which are formed by TMH1 and TMH2 of each of the proteins, LolC and LolE. The triacyl chains and a small N-terminal segment of the lipoprotein were resolved in a vertical arrangement in the upper part of the V-shaped channel. This suggests that the lipoprotein was extracted laterally from the periplasmic side of the IM, as this is also the arrangement of the lipoprotein in the IM. Furthermore, two AMP-PNP-bound LolCDE structures were solved by Tang et al. [151]. Only one of these structures contains a bound lipoprotein. Since AMP-PNP is a non-hydrolyzable ATP analog, these structures represent LolCDE after ATP binding. In the lipoprotein-bound structure, the transporter maintains its V-shaped cavity, and the NBDs are open. In the structure without the lipoprotein, the NBDs are closed. The closure of the NBDs leads to conformational changes in the TMDs, which leads to the closure of the central channel. In particular, TMH1 and TMH2 of LolE are shifted towards LolC. Additionally, the TMH2 of LolE clashes with the triacylcysteine of the lipoprotein of the ligand-bound state, which leads to the extrusion of the lipoprotein.

Sharma and colleagues published cryo-EM structures of LolCDE in a nucleotide-free and a nucleotide-bound state reconstituted in nanodiscs, which represent a native-like lipid environment [152]. In summary, the two determined structures show a very high similarity to the structural counterparts determined by Tang et al. [151]. The nucleotide-bound state was determined using an ADP-vanadate complex trapped in the ATP binding site. In addition, in the vanadate-trapped structure, the two LolD proteins come into close contact, which leads to the uplifting of the TMH2. Thus, ATP-binding leads to the extrusion of the lipoproteins from the TMDs to LolA. Sharma et al. further points out that in this conformation, TMH2 closes the lateral opening between LolC and LolE [152]. This could serve as a further mechanism for preventing substrate entry before the completion of a full transport cycle. In comparison to the structures of Tang et al. [151], the nucleotide-free structure of Sharma et al. [152] shows a significant difference in the binding of the lipoproteins. In the structure of Sharma et al. [152], the N-terminally linked acyl chain of the lipoprotein adopts a rather horizontal conformation in the cavity of LolCDE compared to the rather vertical arrangement of this acyl chain in the structure of Tang et al. [151].

In a relatively recent study, Bei et al. presented cryo-EM structures of nanodisc-reconstituted LolCDE in the apo, lipoprotein-bound, and AMP-PNP-bound states [153]. In general, the structures are highly similar to the corresponding structures published by Tang et al. [151] and Sharma et al. [152]. However, Bei et al. [153] pointed out a remarkable difference in the distances of the periplasmatic domains of LolC and LolE in the apo state compared to the apo state determined by Tang et al. [151]. Nevertheless, the structures of Bei et al. [153] also suggested that the binding of ATP leads to the closure of the central cavity and to the extrusion of the lipoprotein. The three structures determined by Bei et al. [153] are depicted exemplarily for the LolCDE transport mechanism in Figure 11. Panel B shows how the movement of TMH2 of LolE leads to the blockage of the central cavity and, thus, to the extrusion of the lipoprotein.

Putting it all together, the published LolCDE structures in different transport states show a high similarity and correspond well with each other, especially regarding the connection of ATP-binding and release of the substrate to the periplasmic chaperone LolA [152,153]. All studies suggested a similar transport mechanism. In the apo state, the lipoprotein is extracted from the outer leaflet of the IM in an energy-independent manner into the V-shaped cavity inside LolCDE. Through ATP-binding, the NBDs dimerize, which leads to movements in the TMDs, resulting in the shuffling of the lipoprotein to LolA. After the dissociation of ADP and the lipoprotein-LolA complex, the central cavity opens again, and the transporter is primed for the next transport cycle. Despite the vast similarities, slight differences exist between specific structures. There are also differences between the ATPase activities of the different structures. For instance, LolCDE prepared in nanodiscs by Sharma and colleagues shows a several-fold higher activity than LolCDE purified in LMNG by Tang et al. [151,152]. The differences could be attributed to the different detergent and lipid environments in which the structures were solved. This further emphasizes the significance and involvement of lipids in the function of membrane transporters and membrane proteins in general. The ABC transporter LolCDE falls in line with a group of emerging ABC transporters, which show a non-classical transport mechanism. The initial suggestion that LolCDE also follows a mechanotransmission mechanism due to the resemblance to MacB [29] was confirmed by the various discussed LolCDE structures. The study of such ABC transporters with non-conventional transport mechanisms enhances our understanding of living systems and could pave the way for novel drug targets and biotechnological applications.

## 6. Conclusions

Here, we have reviewed four ABC transporters that form the energizing component of four bacterial nanomachineries. We also tried to highlight how the use of ATP deviates from the classic “two-side access” [8] and how these machineries adapted to the particular needs of transporting quite different substrates. Although we have witnessed a tremendous increase in knowledge about the function and structure of the nanomachineries, we have also summarized the obvious questions.

Despite the availability of structural information on the whole MacAB-TolC nanomachinery, detailed insights about substrate binding and recognition are still lacking, including intermediate or substrate-bound structures. Especially the role of ATP hydrolysis in substrate translocation and possible feedback from the putative binding site to the NBDs require further investigations.

Similarly, there are still some intriguing features of the Lpt system, such as the role of LptC during the inactive state or the influence of LptC on the binding of LptB_2_FG to ATP and substrate, that need to be studied in detail towards an in-depth understanding of the Lpt system. Additionally, the oligomeric organization of LptA forming the periplasmic bridge is still unknown.

For the Mla system, the fundamental question regarding the directionality of the transport remains open. The influence of the cell wall on the movement of MlaC is also unknown. Furthermore, the role of a non-functional OMP-MlaA assembly remains elusive.

In terms of the Lol system, the variety of structures is impressive though a detailed functional understanding of the shuttling process between IM and OM remains unclear. This emphasizes the necessity of a combination of all kinds of structural, biochemical, and biophysical data to understand these complex systems in detail.

These questions need to be addressed in the future to fully understand and maybe even exploit the beautiful variety of these different nanomachineries in prokaryotes.

## Figures and Tables

**Figure 1 ijms-24-06227-f001:**
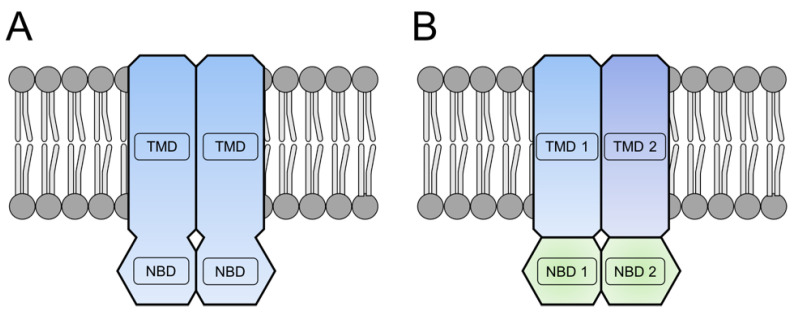
Domain organization of prokaryotic ABC exporters. (**A**)**:** Half-size homodimer. (**B**)**:** Single domain heterodimer. TMD = transmembrane domain. NBD = nucleotide-binding domain. The TMD and NBD are either fused on one polypeptide chain (**A**) or exist as separate polypeptides (**B**).

**Figure 2 ijms-24-06227-f002:**
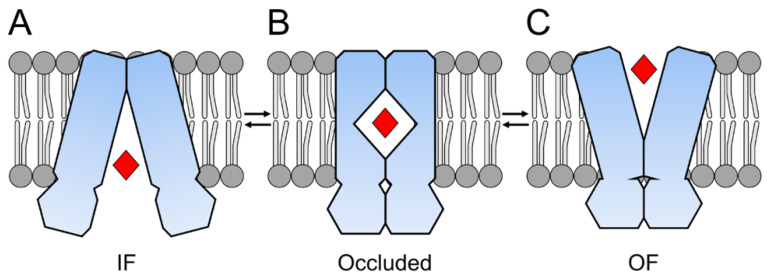
Schematic of a transport cycle of a type IV ABC exporter. (**A**)**:** In the inward-facing (IF) conformation, the transporter binds the substrate. (**B**)**:** During substrate translocation across the membrane, the occluded state is formed with the substrate pocket closed to the inside and outside. (**C**)**:** The transporter adopts the outward-facing (OF) conformation, and the substrate leaves the transporter to the outside.

**Figure 3 ijms-24-06227-f003:**
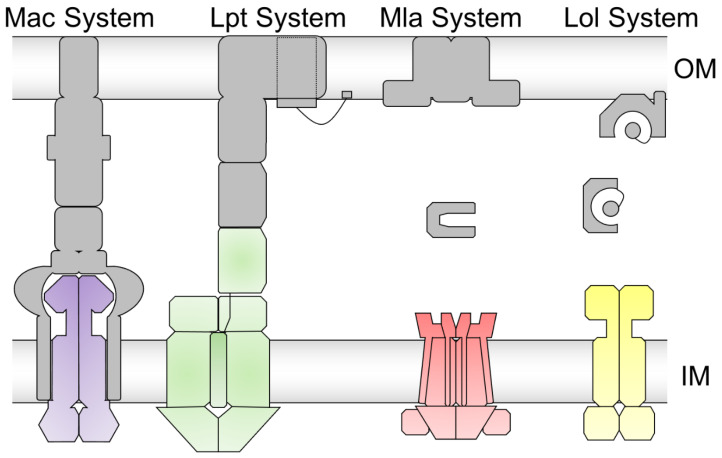
Schematic representation of the Mac, Lpt, Mla, and Lol system. For the Mac system, the ABC transporter MacB is shown in purple. The membrane fusion protein MacA and the outer membrane protein TolC are shown in grey. For the Lpt system, the ABC transporter LptB_2_FGC is shown in green. The periplasmic protein LptA and the translocon in the outer membrane LptDE are shown in grey. For the Mla system, the ABC transporter MlaFEDB is shown in red. The periplasmic protein MlaC and the outer membrane protein MlaA, together with OmpF, are shown in grey. For the Lol system, the ABC transporter LolCDE is shown in yellow. The periplasmic protein LolA and the outer membrane protein LolB are shown in grey. Inner membrane (IM) and outer membrane (OM) are depicted as grey bars. The peptidoglycan layer is omitted for reasons of clarity.

**Figure 4 ijms-24-06227-f004:**
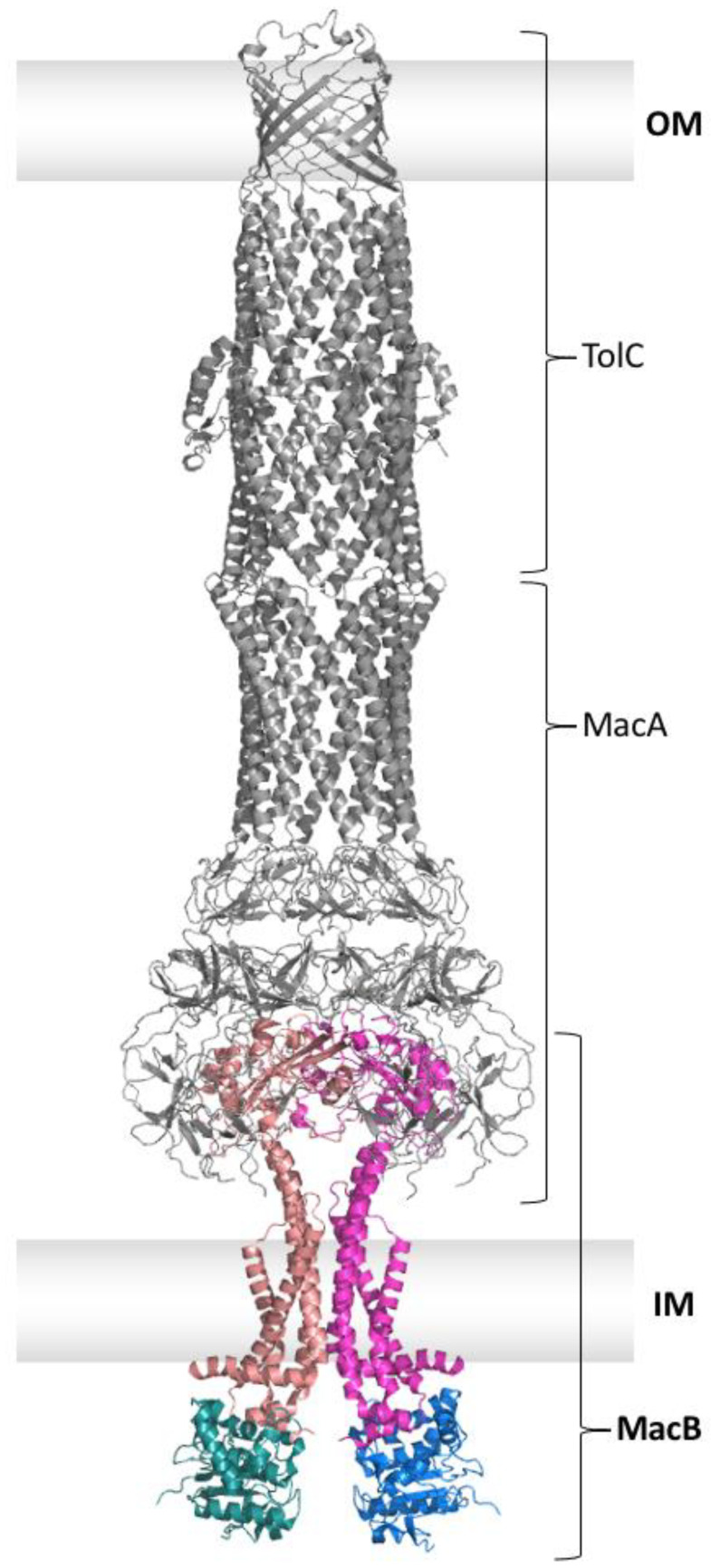
Single particle cryo-EM structure of the assembled MacAB-TolC efflux system (PDB entry 5NIL). Trimeric TolC and hexameric MacA are shown in grey and dimeric MacB in color. The nucleotide-binding domains of MacB are shown in deep teal and marine, and the transmembrane domains, plus the periplasmic part, are shown in salmon and light magenta. The outer membrane (OM) and inner membrane (IM) are displayed as grey boxes. The peptidoglycan layer is omitted for reasons of clarity.

**Figure 5 ijms-24-06227-f005:**
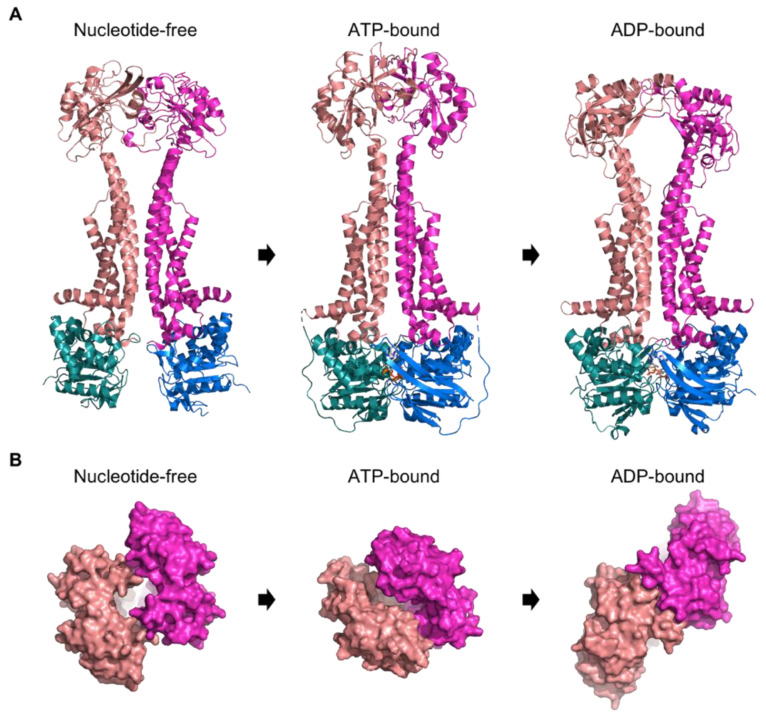
Structures of MacB in a nucleotide-free (PDB entry 5NIL), ATP-bound (PDB entry 5LIL), and ADP-bound (PDB entry 5WS4) state. NBDs are shown in deep teal and marine, and the TMDs with the periplasmic part are shown in salmon and light magenta. (**A**)**:** Side view of MacB in the different nucleotide-free/-bound states. (**B**)**:** Top view of the periplasmic part of MacB in the different nucleotide-free/-bound states.

**Figure 6 ijms-24-06227-f006:**
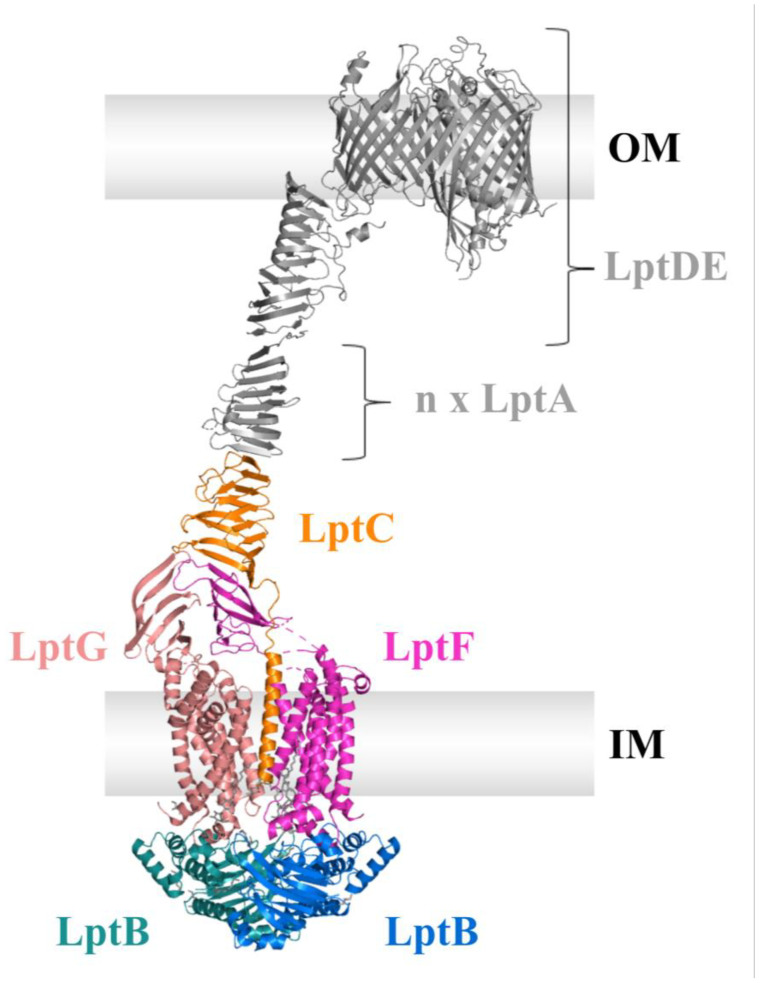
Overview of the Lpt machinery spanning both membranes and the periplasm. Note that the number of LptA oligomers forming the periplasmic bridge is unknown. The figure was created using PyMOL and the PDB entries of LptDE (5IV9, grey in the outer membrane), LptA (2R19, grey in the periplasm), and LptB_2_FGC (6MJP, orange, salmon, light magenta, deep teal and marine in the inner membrane). The outer membrane (OM) and inner membrane (IM) are displayed as grey boxes. The peptidoglycan layer is omitted for reasons of clarity.

**Figure 7 ijms-24-06227-f007:**
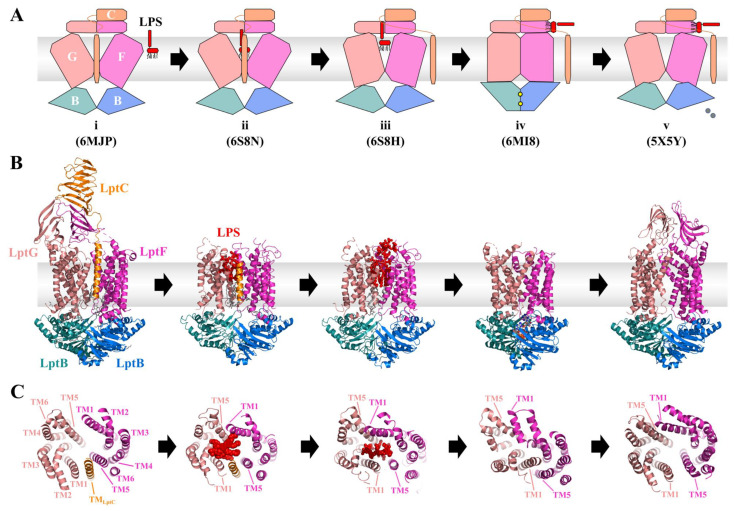
Model and structures of the Lpt systems transport cycle. LptC is shown in orange, while LptF and LptG are shown in light magenta and salmon, respectively. Both LptB protomers are shown in marine and deep teal. LPS is shown in red. Lipids and detergent molecules in the structure are shown as grey sticks, while nucleotides are shown as spheres. (**A**)**:** Schematic model of LPS extraction by the extractor LptB_2_FGC. The different states of the transporter are labeled i-v with the PDB entry for the respective structure. i,ii: LPS enters the LptB_2_FGC cavity from the reader’s side. ii,iii: The LptC TMH leaves the LptFG interface, the cavity tightens and elevates LPS, forming tighter contacts. iii,iv: The LptB_2_ dimer closes, causing the cavity to collapse and pushes LPS upwards to the β-jellyroll domain of LptF. iv,v: ATP (yellow dots) is hydrolyzed to ADP (grey circles) and P_i_, thereby opening LptB_2_ and the cavity for a new extraction cycle. (**B**)**:** Crystal structures and single particle cryo-EM structures according to the different states (**C**)**:** View on the cavity from the periplasmic side (β-jellyroll domains are omitted for clarity). Note that only structures of states ii and iii show LPS. Even though only the structure of state iv shows bound nucleotides, latest data suggest that ATP can bind already during earlier states. Structures of states ii–iv did not resolve the β-jellyroll domain of LptFG. The structure of state ii did not resolve the β-jellyroll domain of LptC, while the structures of states iii–v were lacking LptC completely.

**Figure 8 ijms-24-06227-f008:**
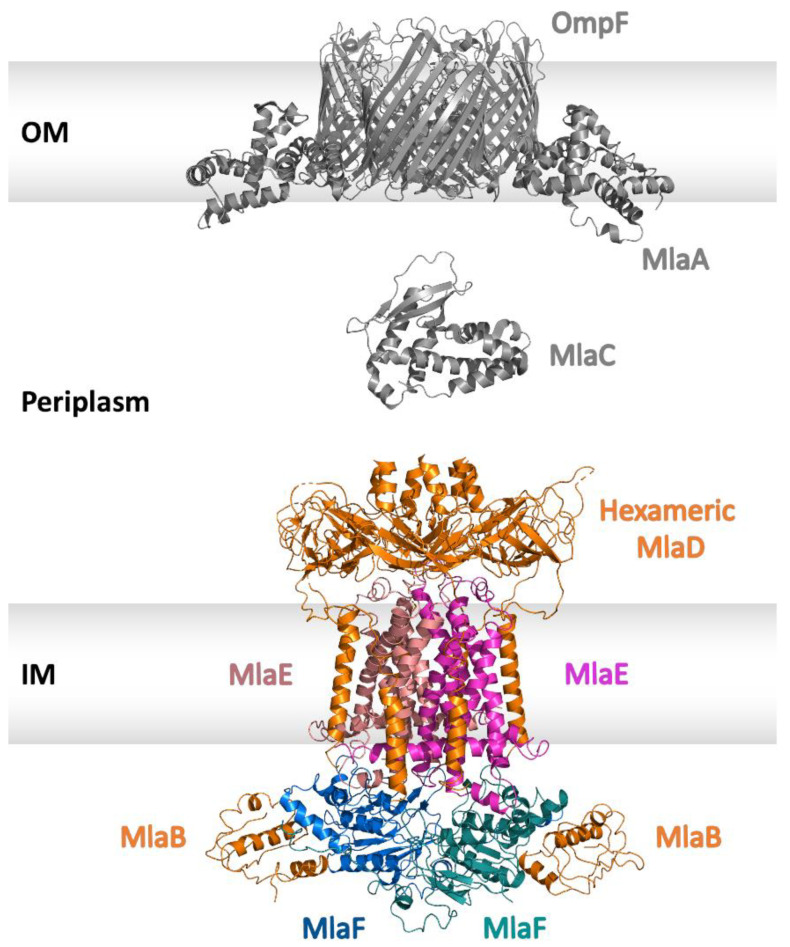
Structural components of the Mla system in a cartoon representation. The homo-trimeric complex of OmpF and MlaA (PDB: 5NUO) situated in the outer membrane is shown in grey color. MlaC (PDB: 6GKI) is shown in the periplasm in grey color. Note: the exact number of MlaC molecules in the periplasm is unknown. The inner membrane complex of MlaFEDB (PDB: 6ZY2) is shown in color: hexameric MlaD, and both MlaB molecules are shown in orange. The two MlaE molecules are shown in salmon and light magenta. The two molecules of MlaF are shown in marine and deep teal. Outer membrane (OM) and inner membrane (IM) are displayed as grey boxes. The peptidoglycan layer is omitted for reasons of clarity.

**Figure 9 ijms-24-06227-f009:**
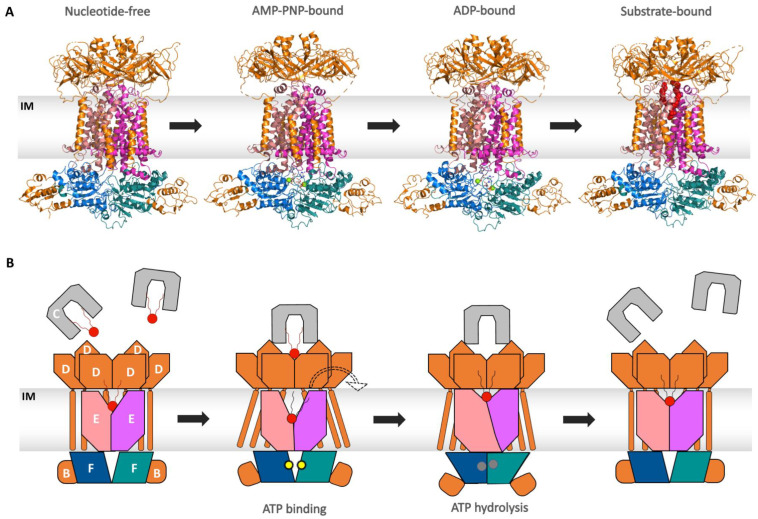
Structures and model representing the transport cycle of the Mla pathway. MlaC is shown in grey. The hexameric MlaD and both MlaB are shown in orange. MlaE is shown in salmon and light magenta. MlaF is shown in marine and deep teal. Lipid molecules are shown in red, and nucleotides are shown as spheres. The IM is displayed as a grey box. The peptidoglycan layer is omitted for reasons of clarity. (**A**)**:** Different states of the ABC transporter during the transport cycle: nucleotide-free (PDB: 6ZY2), AMP-PNP-bound (PDB: 6ZY9), ADP-bound(PDB: 6ZY4), and substrate-bound (PDB: 6ZY3). (**B**)**:** A schematic model of the retrograde transport of the lipid via MlaFEDB. Lipid-loaded MlaC binds to MlaD in the resting state of MlaFEDB. Binding of ATP (yellow dots) prompts the exit of the lipid molecule present in the cavity of MlaE from the last transport cycle. ATP hydrolysis to ADP (grey circles) and P_i_ prompts dimerization of MlaF and conformational changes in MlaE, which ultimately lead to the extraction of lipid from MlaD-MlaC into the cavity of MlaE. Upon release of hydrolyzed products, the conformation gets back to the resting state.

**Figure 10 ijms-24-06227-f010:**
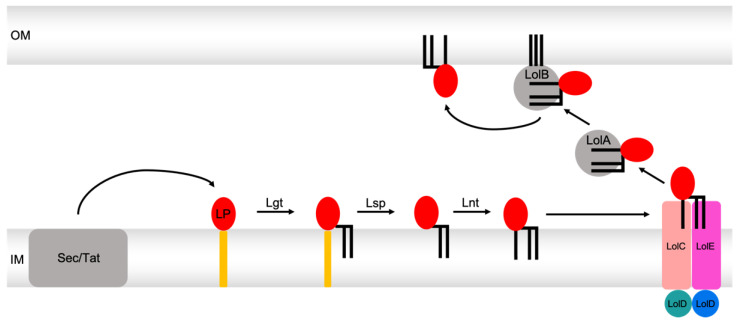
Overview of the Lol pathway. After insertion of lipoproteins (red, labeled one time with LP) via the Sec or the Tat pathway into the IM, they are diacylated (black) by Lgt in a first modification step. This is followed by cleavage of the N-terminal signal sequence (orange) by Lsp. Subsequently, Lnt acylates the newly N-terminally located cysteine, which makes the now mature lipoprotein ready for transport via the Lol machinery. This starts with the extraction of the lipoprotein from the cytoplasmic membrane via the ABC transporter LolCDE (salmon, light magenta, deep teal, and marine), which leads to the delivery of the lipoprotein to the periplasmic chaperone LolA (grey). LolA shuffles the lipoprotein to the last checkpoint LolB (grey), which finally inserts the lipoprotein into the outer membrane.

**Figure 11 ijms-24-06227-f011:**
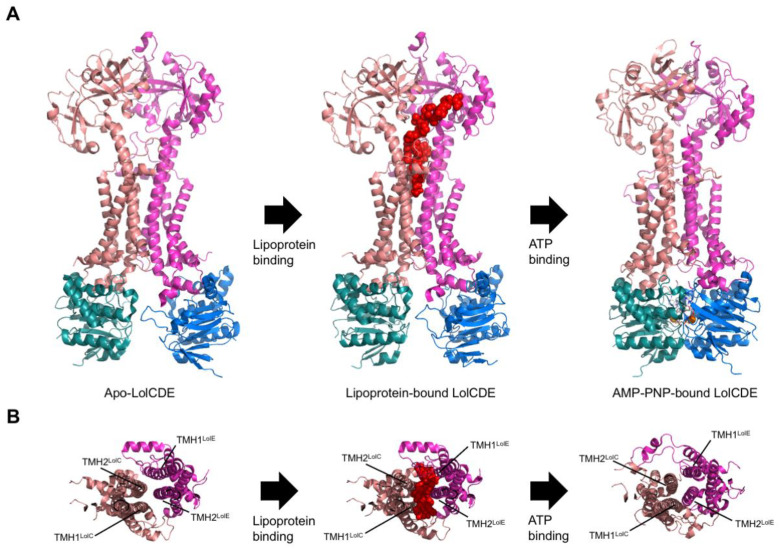
Structures of LolCDE in the Apo (PDB: 7V8M), lipoprotein-bound (PDB: 7V8L), and AMP-PNP-bound (PDB: 7V8I) form. LolC is shown in salmon, LolE is shown in light magenta, one LolD monomer is shown in deep teal, and the other LolD monomer is shown in marine. The lipoprotein is shown in red spheres. (**A**)**:** Side-view of LolCDE. In the Apo state, LolCDE exhibits a V-shaped cavity that the lipoprotein enters from the membrane. Upon ATP binding (AMP-PNP-bound state), the central cavity is closed, and the substrate is shuffled out of LolCDE to LolA. (**B**)**:** Cross-sectional view of the TMDs of LolCDE in the different substrate- or nucleotide-bound states. Movement of TMH2 of LolE upon ATP-binding into the central cavity extrudes the substrate out of the transporter.

## Data Availability

No new data were created or analyzed in this study. Data sharing is not applicable to this article.

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
