# Peer review of "ABC Transporters in Bacterial Nanomachineries"

_ijms, 2023, doi:10.3390/ijms24076227_

Round 1

Reviewer 1 Report

Synopsis:

ABC transporters are key components in several nanomachineries on the plasma membranes of bacteria. They are associated with antibiotic resistance, bacterial membrane biosynthesis, nutrient transport, etc. In recent years, major technical advancement in either X-ray crystallography or cryo-EM has permitted discoveries of high-resolution structures of these bacterial ABC transporters. Consequently, many mechanistic models are proposed based on these structure-function studies. In this review, Bilsing et al discussed in good details about four bacterial secretory nanomachineries, including Mac, Lpt, Mla, and Lol. The manuscript also includes very informative figures that provide a great graphic abstract for the topic. While this reviewer appreciates the wealth of knowledge that the authors delved into each system, the manuscript would greatly benefit from some structural reorganization and rigorous proofreading. As described below, this reviewer provides some suggestions for the authors’ consideration.

Concerns:

1. The Mla system covers subheadings 5-9, while the other three systems were collected under one subheading per system. To make it clear to the audience, this reviewer suggests merging Mla-related sections into one, i.e., 3: Mac, 4: Lpt, 5: Mla, and 6: Lol, and the manuscript ends with 7: conclusion.

2. “Figures” are not cited in the text. Therefore, it reads broken when trying to connect the text to figures.

3. Several citations are missing or formatted incorrectly.

4. Given four big sessions being written, the conclusion paragraph is quite short and in this reviewer’s opinion, does not convey key messages that this manuscript intends to bring up. While each systems I includes some questions and hypotheses, in the conclusion section, this reviewer suggests the authors to reiterate the key questions here, including what’s addressed and what’s not by available structural and functional data.

5. On Line 361, it says Table 6; however, this reviewer can not see any table in this manuscript. Are there tables?

Author Response

Reviewer 1:

Open Review

Quality of English Language

( ) English very difficult to understand/incomprehensible
( ) Extensive editing of English language and style required
( ) Moderate English changes required
(x) English language and style are fine/minor spell check required
( ) I am not qualified to assess the quality of English in this paper

Comments and Suggestions for Authors

Synopsis:

ABC transporters are key components in several nanomachineries on the plasma membranes of bacteria. They are associated with antibiotic resistance, bacterial membrane biosynthesis, nutrient transport, etc. In recent years, major technical advancement in either X-ray crystallography or cryo-EM has permitted discoveries of high-resolution structures of these bacterial ABC transporters. Consequently, many mechanistic models are proposed based on these structure-function studies. In this review, Bilsing et al discussed in good details about four bacterial secretory nanomachineries, including Mac, Lpt, Mla, and Lol. The manuscript also includes very informative figures that provide a great graphic abstract for the topic. While this reviewer appreciates the wealth of knowledge that the authors delved into each system, the manuscript would greatly benefit from some structural reorganization and rigorous proofreading. As described below, this reviewer provides some suggestions for the authors’ consideration.

Concerns:

  1. The Mla system covers subheadings 5-9, while the other three systems were collected under one subheading per system. To make it clear to the audience, this reviewer suggests merging Mla-related sections into one, i.e., 3: Mac, 4: Lpt, 5: Mla, and 6: Lol, and the manuscript ends with 7: conclusion.

The sections have been changed and renumbered as suggested by the reviewer.

  1. “Figures” are not cited in the text. Therefore, it reads broken when trying to connect the text to figures.

All figures have been properly cited now throughout the text of the revised version.

  1. Several citations are missing or formatted incorrectly.

All the references have been thoroughly checked and formatted.

  1. Given four big sessions being written, the conclusion paragraph is quite short and in this reviewer’s opinion, does not convey key messages that this manuscript intends to bring up. While each systems I includes some questions and hypotheses, in the conclusion section, this reviewer suggests the authors to reiterate the key questions here, including what’s addressed and what’s not by available structural and functional data.

The authors have expanded the conclusion section as suggested by the reviewer.

  1. On Line 361, it says Table 6; however, this reviewer can not see any table in this manuscript. Are there tables?

There are no tables in the manuscript and it was a typo. We apologize for this.

Reviewer 2 Report

The manuscript presents a review of four systems in Gram-negative bacteria which function as transport proteins at a membrane level and function with energy from ATP hydrolysis: Mac system related to drug resistance, the Lpt systems catalyzing vectorial LPS transport, the Mla systems responsible for phospholipid transport, and the Lol system, required for lipoprotein transport. The review explains the function of the proteins of interest very deeply and with well explained relation to their structure. Information about the limitations of the knowledge about the function of the reviewed proteins will help in further research. It can be accepted for publication. Few editorial remarks have to be taken into account.

Figures 3 and 6 were cited in the text but others – not. Please, revise! In general, the figures are clear and shed more light on the function of these transporters, therefore is very important to cite them at a suitable place.

Line 160; Line 167: Please, pay attention to the reported error.

Lines 231-267: Please, pay attention to the reported errors. Similar errors could be found in the entire manuscript and they require attention.

Line 483: Tang et al. – Please, cite the reference correctly.

Author Response

Reviewer 2:

Open Review

Quality of English Language

( ) English very difficult to understand/incomprehensible
( ) Extensive editing of English language and style required
( ) Moderate English changes required
(x) English language and style are fine/minor spell check required
( ) I am not qualified to assess the quality of English in this paper

Comments and Suggestions for Authors

The manuscript presents a review of four systems in Gram-negative bacteria which function as transport proteins at a membrane level and function with energy from ATP hydrolysis: Mac system related to drug resistance, the Lpt systems catalyzing vectorial LPS transport, the Mla systems responsible for phospholipid transport, and the Lol system, required for lipoprotein transport. The review explains the function of the proteins of interest very deeply and with well explained relation to their structure. Information about the limitations of the knowledge about the function of the reviewed proteins will help in further research. It can be accepted for publication. Few editorial remarks have to be taken into account.

Figures 3 and 6 were cited in the text but others – not. Please, revise! In general, the figures are clear and shed more light on the function of these transporters, therefore is very important to cite them at a suitable place.

All the figure citations have been revised throughout the manuscript.

Line 160; Line 167: Please, pay attention to the reported error.

The reported error has been rectified and the figures have been cited accordingly.

Lines 231-267: Please, pay attention to the reported errors. Similar errors could be found in the entire manuscript and they require attention.

As suggested by the reviewer, all the reported errors have been checked and rectified.

Line 483: Tang et al. – Please, cite the reference correctly.

The reference has been cited correctly now.

Round 2

Reviewer 1 Report

The authors have addressed all concerns.